# The "Irish" Female Servant in Valerie Martin's *Mary Reilly* and Elaine Bergstrom's *Blood to Blood*

**Dara Downey**

School of English, Arts Building, Trinity College Dublin, Dublin 2, Ireland; downeyd@tcd.ie

**Abstract:** This article examines two neo-Victorian novels by American writers—Valerie Martin's *Mary Reilly* (1990) and Elaine Bergstrom's *Blood to Blood* (2000)—which "write back" to Robert Louis Stevenson's *Strange Case of Dr Jekyll and Mr Hyde* (1886) and Bram Stoker's *Dracula* (1897), respectively. Both novels ostensibly critique the socio-cultural inequalities of Victorian London, particularly for women, immigrants, and the working class, and the gender and class politics and structures of the original texts. However, as this article demonstrates, the presence of invented Irish female servants as key figures in these "re-visionary" narratives also undermines some aspects of this critique. Despite acting as gothic heroines, figures who traditionally uncover patriarchal abuses, these servant characters also facilitate their employers' lives and negotiations of the supernatural (with varying degrees of success), while also themselves becoming associated with gothic monstrosity, via their extended associations with Irish-Catholic violence and barbarity on both sides of the Atlantic. This article therefore argues that Irish servant figures in neo-Victorian texts by American writers function as complex signifiers of pastness and barbarity, but also of assimilation and progressive modernization. Indeed, the more "Irish" the servant, the better equipped she will be to help her employer navigate the world of the supernatural.

**Keywords:** servants; Irish; emigration; immigration; domestic; gothic; historiographical metafiction; re-visionary fiction; *Dracula*; *Strange Case of Dr Jekyll and Mr Hyde*

## 1. Introduction

In Valerie Martin's 1990 novel *Mary Reilly*, the eponymous narrator, a servant working in the household of Dr Henry Jekyll (a version of the character from Robert Louis Stevenson's *Strange Case of Dr Jekyll and Mr Hyde* (Stevenson [1886] 2008), henceforth referred to as "*Jekyll and Hyde*") finds herself sorely puzzled by Jekyll's increasingly troubling behavior. Mary is an enthusiastic cleaner, finding personal satisfaction in working for a "Master" who she believes (in the face of mounting evidence to the contrary) to be morally upright and kind. After a particularly perplexing series of events, she tells the reader, "I had a fancy [ . . . ] that I would take every lamp in the house and light them in one room and then, if I stood among them, I might see properly" (Martin 2003, p. 190). This image is qualified later on when, after Edward Hyde's body has been found in Jekyll's private "cabinet" and there is no sign of her employer, Mary "held up the lamp so that [she] might see farther, but [her] own shadow fell across the table" (Martin 2003, p. 235). What this, and her continued failure to "see properly" that Jekyll and Hyde are essentially the same person, implies is that she is standing in her own light. She is, in other words, part of the reason why she has been (and remains throughout the narrative) unable to see clearly what is going on in a house that she cleans so exhaustively.

This image establishes a number of motifs that, as I argue in this article, are central to understanding the depiction of Mary and her position in Jekyll's house and Martin's novel. First and foremost, her use of a lamp as both a literal source of light and a metaphor for moral and epistemological illumination positions Mary as a gothic heroine, a direct descendent of Ann Radcliffe's trembling-yet-inquisitive

protagonists. In Martin's novel, however, the gothic-heroine role is complicated by Mary's implied Irishness because, as this article argues, both the Irish and servant figures have long functioned (in a set of homogenizing discourses that tend to bulldoze over complexities and diversities in relation to class and creed) as signifiers of a gothic past that has been surmounted by their middle-class, Anglo-Protestant employers. It is for this reason, I argue, that Mary is also depicted here as experiencing a form of doubling (in the sense of being obstructed by her own shadow), and it is the combination of this self-haunting and her unwavering devotion to Jekyll that, I argue, impedes her gothic quest to uncover the dark secrets that render her workplace—and her employer—haunted and dangerous. Both that doubling and her status as a failed gothic heroine can be attributed to the novel's ambivalence in positioning Mary as Irish (made explicit in the 1996 Stephen Frears film adaptation). Mary is figured as an extension or "prosthesis" (Robbins 1986, p. 153; Thompson 2015, p. 91) of her "Master," her role as servant entangling her in the concealment of his secrets. Simultaneously, however, she is paralleled with *his* double, Hyde, who can be read as an avatar of the Irish terrorist in the *fin-de-siècle* imagination, carrying gothicized ethnic associations that resonate throughout Martin's novel in ways that highlight the persistence of negative stereotypes of Irishness in contemporary American-gothic fiction.

Mary Reilly's ambiguous Irish origins can, I argue, be usefully illuminated by reading Martin's novel in parallel with another late-twentieth-century rewriting of a classic late-nineteenth-century gothic text: Elaine Bergstrom's *Blood to Blood* (Bergstrom 2000), a sequel to *Mina* (Bergstrom 1994, published under the pseudonym Marie Kiraly), which co-opts and extends the narrative world of Bram Stoker's *Dracula* (Stoker [1897] 1993), focusing on Mina's experiences in the aftermath of the original, and which also features an Irish female servant named Colleen. Bergstrom's Colleen is, unlike Mary, explicitly depicted as Irish. Reading *Mary Reilly* in tandem with *Blood to Blood* helps highlight the ways in which Irish servants on both sides of the Atlantic continue to be associated with a gothicized European past, an association that can be traced back to the Victorian source material, and indeed further back to the eighteenth-century gothic novel. Both *Mary Reilly* and *Blood to Blood* ostensibly critique the socio-cultural inequalities of Victorian London, which rendered the city particularly perilous for those who fit the stereotype of the Irish domestic servant as female, a recent immigrant, Catholic, and working class. However, as Mary's failure or refusal to uncover Jekyll's terrible secrets in time to save him illustrates, both texts also exploit these very inequalities and dangers to produce their gothic effects, fetishizing the benighted past against which they establish their supposedly more progressive modernity.

In order to elucidate how this works, I begin by exploring the novels' status as "re-visionary" works that repeat and rewrite canonical Victorian-gothic texts (see Domsch 2012, pp. 105–06). As I argue, the neo-Victorian gothic's use of Victorian London as a privileged site of pastness mirrors the gothic heroine's confrontation with a historical moment that must be exposed and exorcized to make way for a more "enlightened" present. This leads into a discussion of domestic service, which, seen in nineteenth-century America as fundamentally archaic and even un-American, can be understood within a similar historical-gothic framework. Exacerbating this perception was the fact that domestic service was dominated in nineteenth-century America by young, Irish-born, Catholic women, who were themselves seen as uncivilized, archaic, and un-American—essentially, as gothic. Examining these issues highlights the various layers of figurative pastness that continue to accrue to fictional Irish servant girls in Martin and Bergstrom's novels, and the extent to which they sit uneasily with their perceived capacity for cultural assimilation and "modernization" through service. Both novels rely on these associations (which reify the fluid historical intersections of ethnicity, religion, and class), even as they claim to raise from invisibility those who were the subject of such stereotyping.

## 2. Re-Visionary Historical Gothic

*Mary Reilly* and *Blood to Blood* are examples of what I term "re-visionary historical gothic." These are twentieth-century novels by American women that confer active roles upon female characters who are underrepresented or silenced in *Dracula* and *Jekyll and Hyde*, which are dominated by middle-



and upper-class male characters. Both of Kiraly/Bergstrom's novels begin by announcing their close association with Stoker's *Dracula*, before moving beyond it to explore a range of issues that may resonate with late-twentieth-century readers, including the struggles of nascent feminism (fought in the workplace and the bedroom), social inequality, and the treatment of immigrants. The first book, *Mina*, takes up the journals and letters of its eponymous heroine from the moment at which they are abruptly cut off in *Dracula*. Sensing that the Count's blood still flows through her veins even after his apparent destruction, this new version of Mina is compelled to break free from the shackles of Victorian femininity and her stifling marriage to Jonathan Harker by becoming better acquainted with both vampirism (see Auerbach 1995, p. 148; Creed 2005, p. 72) and the seedier elements of London life. The sequel, *Blood to Blood*, published under Bergstrom's own name, continues but also splits off from Mina's narration, following Dracula's half-sister Joanna's journey to England, and her efforts to overcome the traumas and isolation of her very long life. Joanna is aided by Colleen Kelley O'Shaunnasy, a London-Irish runaway, who quickly becomes the vampire's devoted handmaiden and most reliable source of nourishment, a practice which soon compromises Colleen's humanity. Joanna also claims her brother's properties in London, and thus becomes entangled with the Harkers, as well as beginning a perilous liaison with Arthur Holmwood, who is haunted by the memory of staking his fiancée, Lucy, and beginning to question Abraham Van Helsing's authority. *Blood to Blood*, which revels in the moral and literal murkiness of late-Victorian London, ends with Colleen (now a vampire) almost killing Dr Rhys, a philanthropic doctor who befriends Mina but who, it emerges, is in fact a serial killer (the novel's positioning of its only Hindu character as a compulsive murderer is one of its more uncomfortable aspects). Mina shoots Dr Rhys before Colleen can drain him, saving her from transforming completely into a murderous monster. Colleen is therefore left in a liminal state between human and vampire, while continuing to work as Joanna's servant—part of Mina's diverse group of female friends, but nonetheless marginalized by her class and undead status, identity markers that, as discussed below, are depicted as inextricable here and elsewhere.

Unlike Colleen, who is entirely Bergstrom's invention, Mary Reilly, the first-person eponymous narrator of Martin's novel, can be identified within the silences in Stevenson's original. Near the crisis point in *Jekyll and Hyde*, Gabriel Utterson, a lawyer and close friend of Dr Jekyll who begins to suspect that something is horribly wrong in the Jekyll household, is summoned there by the most senior servant, Poole. The two men are greeted by the entire staff, who

> [ . . . ] stood huddled together like a flock of sheep. At the sight of Mr Utterson, the housemaid broke into hysterical whimpering; and the cook, crying out "Bless God! It's Mr Utterson," ran forward as if to take him in her arms.
>
> [ . . . ] "They're all afraid," said Poole.
>
> Blank silence followed, no one protesting; only the maid lifted up her voice and now wept loudly.
>
> "Hold your tongue!" Poole said to her with a ferocity of accent that testified to his jangled nerves [ . . . ]. (Stevenson [1886] 2008, p. 35)

This crying housemaid would appear to be the "original" of Martin's Mary Reilly. While the servants clearly have a better sense of what is going on in the house than the male authority figures, she and the rest of the staff remain peripheral to the final revelation. This idea is capitalized upon by Martin's novel; as she cleans and tidies the house, Mary is privy to what occurs "off stage" in Stevenson's novella, witnessing first hand Dr Jekyll's uncharacteristic mood swings, and the considerable liberties taken by the repulsive Hyde, who is passed off as an assistant in Jekyll's scientific researches, but who acts like an overprivileged family member, simultaneously shambling and strutting. Furthermore, as Bryk (2004) asserts, endowing the crying maid silenced by Poole "with the narrative voice clearly constitutes an 'empowering act' [ . . . ] an act of historical reparation and an attempt at subverting

the political bias of the original text" (p. 205). Saxey (2009) similarly sees *Mary Reilly* as part of a late-twentieth-century literary impulse to re-imagine the Victorian past by critiquing and rejecting its repressions and injustices, which the twentieth century has, it is implied, successfully surmounted (pp. 58–61; see also Heilmann and Llewellyn 2007, pp. 2–3, 8; De Groot 2010, pp. 67–68).

This is, in fact, the central narrative aim of a vast array of contemporary fictional texts that, like *Mary Reilly* and Kiraly/Bergstrom's books, "resurrect" Victorian literary monsters (Domsch 2012, p. 102). Widdowson (2006) refers to such texts as "re-visionary fiction"—texts that "'write back to'–indeed, 'rewrite'–canonic [*sic*] texts from the past" (p. 491). Specifically, re-visionary fiction works to restore "a voice, a history and an identity to those hitherto exploited, marginalized and silenced by dominant interests and ideologies" (Widdowson 2006, pp. 505–06; see also Robinson 2011, p. 19; Duperray 2012, p. 191; Hutcheon and O'Flynn 2013, pp. 171–72). In the case of *Mary Reilly* and *Blood to Blood*, it is the voices of domestic servants that are restored in this way, in an effort to undo their socio-cultural marginalization (Katzman 1981, pp. 16–17; Thompson 2015, p. 91). What is more, Martin and Bergstrom's novels "re-vision" works—*Jekyll and Hyde* and *Dracula*—that themselves write back to and alter earlier gothic configurations, by taking the central investigative role away from a young, imperiled heroine and bestowing it instead upon a homosocial group of privileged men (see Sedgwick 1992)—that is, Jekyll's friends Utterson and Lanyon, and in *Dracula*, the all-male "Crew of Light" (see Craft 1984, p. 109).

Specifically, by exploring the castle, mansion, abbey, or ruin when she has been warned not to (and in which she is frequently imprisoned), thereby exposing the villain's horrible deeds, the heroine in much eighteenth-century gothic succeeds in reinstating (an implicitly middle-class, broadly Protestant conception of) normality and safety (see Ellis 1989; DeLamotte 1990; Gillespie 2014, p. 62). This gothic quest therefore entails shedding literal and figurative light upon the dark places occupied by tyranny, superstition, and the exploitation of both by representatives of patriarchal power. Consequently, lamps, candles, and guttering tapers fitfully illuminate the gloomy world of, for example, Radcliffe's *The Mysteries of Udolpho* (Radcliffe [1794] 2001), J. Sheridan Le Fanu's *Uncle Silas* (Le Fanu [1864] 1988), and Louisa May Alcott's *A Whisper in the Dark* (Alcott [1863] 1889), a motif that *Mary Reilly* repeats and problematizes. As the heroine "look[s] back into the past to establish a true history which the patriarch had sought to falsify" by concealing his nefarious deeds (Milbank 1992, p. 11), she both illuminates and rebels against this obsolete system, enacting an exorcism/house cleaning that breaks the past's choke-hold on the present (pp. 10–12).

Moreover, while the "present" of the gothic is haunted by a hidden past, that "present" is already itself situated in the distant past. The heroine's struggles generally take place in a time (pre-Reformation) and place (continental Europe) far removed from that of the author, suggesting that the abuses that she uncovers cannot possibly take place in the here and now (see Ellis 1989, p. 51; Clery 2000, p. 14; Haggerty 2007, p. 35; De Groot 2010, p. 16). This is also precisely what much neo-Victorian fiction does, employing gothic's historiographical separation of secure present from benighted past (Kohlke and Gutleben 2012, p. 4). It frequently does so by making the Victorian era to the present day what the "Middle Ages" were to the eighteenth-century gothicists (Kohlke and Gutleben 2012, pp. 10–12; Davies 2015, p. 9; Brindle 2012, p. 296; Muller 2012, p. 122), and, in particular, Victorian London is obsessively re-imagined as a site of gothic dread (see Robinson 2011, p. 150ff). Heilmann and Llewellyn (2010) attribute this trope to a desire to recreate the perceived wholeness and coherence of a Victorian "canon" (p. 11; see also Widdowson 2006, p. 491). Such fiction is implicitly invested in keeping the Victorian status quo alive, and therefore contains nostalgic, conservative tendencies that undermine its supposedly progressive aims (Heilmann and Llewellyn 2010, p. 6).

## 3. The Irish Servant

In Martin and Bergstrom's re-visionary historical gothic, these nostalgic impulses are two-fold, referencing both *fin-de-siècle* London and the American nineteenth century. On the one hand, Mary and Colleen function as gothic heroines who take the power back from Stevenson's and Stoker's

male characters, while simultaneously redressing the invisibility of the working class in canonical Victorian gothic. On the other hand, both women serve to allegorize the integration of Irish servant girls into "respectable" American society, but remain haunted by the gothic reputation of the Irish in nineteenth-century Britain, a reputation that manifests as a form of doubling. It was negative attitudes towards Catholicism (perceived as superstition) on the part of the Anglo-Protestant middle class (a diverse group united by their self-identification in opposition to Catholicism) that facilitated the positioning of the Irish as gothic monsters in both Britain and the United States. As Jarlath Killeen points out, from at least the seventeenth century, English colonial discourse figured the Irish as backward, bestial, and subhuman, and the country itself as inhabiting an earlier point in history than its neighbor (Killeen 2014, pp. 3, 5, 9–10). Such depictions formed the basis for much Anglo-Irish gothic from the late eighteenth century onwards. Arguably reaching its apogee in the works of writers such as Le Fanu, Stoker, and Oscar Wilde, this was primarily (though by no means exclusively) a Protestant form, in which Catholic figures are depicted as terrifying monsters threatening an imperiled Anglican minority (Killeen 2014, p. 42; see also Eagleton 1995, pp. 188–89; Sage 2000, p. 86; Gillespie 2014; Morin 2014). However, as Shanahan (2014) suggests, a central source of anxiety in Irish-Protestant gothic (a mode that encompasses writers from a range of denominations) is the half-acknowledged, guilty realization that this monstrosity is the direct consequence of British political interventions in Ireland. The Anglo-Protestant community in Ireland therefore contained within itself the seeds of an equally terrifying monstrosity, one born of complicity with tyrannical power (Shanahan 2014, p. 85), and that destabilizes and highlights the artificiality of neat oppositions between Catholic "savages" and beleaguered Protestants. This mirroring between two sides of the socio-religious divide appears as "a state akin to that described by Monçada in *Melmoth* [*the Wanderer*, by Charles Maturin] as a 'haunting of yourself by your own spectre, while you still live'" (Shanahan 2014, p. 85, quoting from Maturin [1820] 2000, p. 263).

If this definition is extended to include *Jekyll and Hyde* as an example of Scottish-Protestant gothic (again, with "Protestant" here taken to denote "not-Catholic"), both Stevenson's novella and Stoker's *Dracula* emerge as texts where, like Wilde's Dorian Gray, the British protagonists are subject to a horrifying doubling, haunted by their own Irish-Catholic shadows (Duncan 2000, pp. 70–71; Davidson 2018, p. 34). *Jekyll and Hyde* and *Dracula* are both set in London, but gesture towards the more peripheral origins of their authors (despite their very different Protestant affiliations), in the form of the Catholic paraphernalia such as crosses and communion wafer that Van Helsing rightly insists are the only weapons that can hinder Count Dracula; and in the figurative associations of Hyde with Irish immigrants and republican violence, issues that had particular resonances in nineteenth-century Scotland. As Emily A. Jackson (2013) points out, Irishness "was often viewed through the lens of the Victorian preoccupation with theories of degeneration," which suggested that "the Irish were not simply inferior to other British citizenry; they were actually less evolved" (Jackson 2013, p. 79). In particular, nineteenth-century English discourse frequently connected Irish-Catholic "superstition" with Fenian political violence (Hoeveler 2014, p. 39; Shanahan 2014, p. 84), which in turn was associated with "'the worst acts of mediaeval tyrants and of savage tribes'" (Herman 2017, p. 3, quoting from The Times, 16 January 1885, p. 9)—that is, with the pre-Enlightenment "barbarism" that gothic heroines resist and depose (Morin 2014, pp. 16–17). This finds echoes in Stevenson's novella; "[i]n the British press, the Irish were known for their vicious brutality," and "particularly for [ . . . ] stamping and trampling" (Jackson 2013, pp. 78–79), the first violent act that Utterson sees Hyde committing. Consequently, a number of critics have read Hyde as an analogue of the Irish Fenian "hooligan" (Brantlinger and Boyle 1988, p. 273; Brantlinger 2010, p. 61; Jackson 2013, p. 78).

This reading is given weight by Jackson's argument that many Scots sought to identify as British (rather than "Celtic"), in opposition to the allegedly violent, backward Irish (Jackson 2013, pp. 78–79). The Irish were unwelcome but necessary intruders, as the British Empire relied upon them to perform the physically demanding and menial tasks undergirding its modernity. Jackson therefore sees *Jekyll and Hyde* as "an allegory not of colonial fear and the desire for expulsion, but of colonial fear and

the necessity for integration" (Jackson 2013, p. 81), as Jekyll mistakenly attempts to free himself from the less desirable elements of his personality by siphoning them off into Hyde (see Stevenson [1886] 2008, pp. 52–57), a process that ultimately leads to his physical and psychological *dis*integration, a disintegration that others have read as metonymizing Scotland's uneasy relationship to its recent past, or as a literalization of Calvinist self-scrutiny (Duncan 2000, p. 78; Sage 2000, p. 90; Stout 2010, p. 551; Milbank 2018, p. 98). In Jackson's reading, Britain was more secure if the Irish could be assimilated rather than othered. What this means, however, is that the Scottish, like the Protestant (Anglo-)Irish, were haunted by the presence of a primitive Irish-Catholic population from whom they tried to distinguish themselves, precisely because of the discursive and ethnic proximity between the two populations (Jackson 2013, pp. 78–79). In Stevenson's *The Dynamiter* ([1885] 1906, with Fanny Van de Grift Stevenson), for example, the English character Somerset, drafted to join a group of Fenian terrorists, expresses his unease with the violence they employ in terms that closely echo Jekyll's relationship with Hyde; Somerset is troubled by a "sense of something evil, irregular, and underhand, [that] haunted and depressed him" (Stevenson and Stevenson 1906, p. 427). Stevenson's novella and *Dracula* therefore both betray a horrified fascination with Catholic and Celtic "barbarism," while working hard to define a "proper," unified British identity in opposition to it (Hanson 1997, p. 25; Roden 2007, p. 13; Haggerty 2007; Killeen 2014, p. 34; Morin 2014, pp. 19–20). Irishness was the dark double of nineteenth-century British identity, employed in such depictions more as a haunting idea than a historical reality.

In the American context, the immigrant Irish, and particularly Irish servants, were similarly portrayed as invading others imperiling nineteenth-century middle-class Protestant identity, but also as central to the construction of that identity. Hiring servants was a signifier of material success, and permitted women to free themselves from domestic drudgery to attend to "higher callings" such as philanthropy and social reform (Sutherland 1981, pp. 11–13; Katzman 1981, pp. 165–67; O'Neill 2009, p. 116; Turner 2012, pp. 63, 101), as Mina does in *Blood to Blood*, setting up a charitable institution for working women, from which she hires her servant, Essie. Servants were also the invisible hands permitting the construction of the myth of the "angel in the house," who was charged with keeping the home physically and morally spotless for husband and children (Flynn 2011, p. 4). However, for many contemporary commentators, there was something essentially un-American about the practice (see Dudden, p. 124)—something out of keeping with the nation's self-defining rhetoric. This rhetoric was founded upon much the same Enlightenment ideals as the eighteenth-century gothic and offered a spatio-temporal model of civilization that situated barbarism in both the past and in Europe. As Hector St John de Crevecoeur put it in 1782, "[h]e is an American, who leaving behind him all his ancient prejudices and manners, receives new ones from the new mode of life he has embraced" (De Crevecoeur [1782] 1904, p. 54). To go to America, in these terms, is to start anew, to access opportunities unavailable in the more hide-bound "Old World." To go to America and to enter domestic service, however, was to move in two directions at once—towards both the democratic future and the "feudal" past, signified by the "medieval," anti-democratic, archaic, and anachronistic institution of domestic servitude (Turner 2012, p. 92; Sutherland 1981, pp. 5, 102, 121–23, 158; Katzman 1981, pp. 95, 107, 146, 240–42). As Harriet Beecher Stowe asserted in 1865, while America harbored "no entailed property, [ ... ] no hereditary titles, no monopolies, no privileged classes," domestic service "still retain[ed] about it something of the influences from feudal times" (Stowe [1865] 1999, pp. 499–500).

Conveniently, however, the great age of domestic service coincided with and was facilitated by an influx of immigrants, particularly from Ireland, around the middle of the nineteenth century, as a result of the Great Famine (1845–1849). This helped ease Americans' cognitive dissonance regarding domestic service, as perceived ethnic, racial, and socio-cultural differences allowed Irish and (following the American Civil War) African-American servants to be discursively dehumanized to the extent that their subjection within such an anti-democratic institution seemed all but inevitable (Sutherland 1981, p. 41). In the Northern states, Irish women dominated both the labor market for domestic service and popular perceptions of servants throughout the nineteenth century (Howes 2009, p. 98). The stereotype

persisted even when second-generation Irish began to shun service and were replaced by African Americans in the early decades of the twentieth century; despite the statistics, the archetypal domestic servant remained a young, unmarried Irish girl (Sutherland 1981, pp. 50, 59, 183; Katzman 1981, pp. 66, 273). These popular perceptions were by no means always positive. The Irish were seen as part of the "dangerous classes" (Dezell 2001, p. 153), and prospective servants were described by one employer as "hideous Irish monsters" not "fit to enter a decent house" (Fisher 1967, p. 558). The Irish women who came to work in "respectable" Protestant, middle-class households were caricatured in cartoons and satirical fiction as "terrible cooks, poor house cleaners, temperamental if not violent, and clumsy and awkward in handling the family's precious china and crockery" (Diner 1983, p. 86; Dudden 1983, p. 66). More generally, the Irish were stereotyped "as miserably poor, addicted to drink and fiddle music, and tied to deplorably primitive farming methods," as well as being "superstitious; believing in fairy thorns, little people, and the menace of the spirit world at Halloween" (Rodgers 2009, p. 33; see also McDannell 1994, pp. 13–14; Murphy 2000, p. 155). This was no small matter in a country that associated its enlightened, democratic status with the Protestant affiliations of the majority of its population, despite the vast diversity within Protestantism itself (Hill 2007, p. 98; Lynch-Brennan 2009, p. 61). Catholicism was, by contrast, seen as stuck in the past, capricious, irrational, and hence barbaric (Flynn 2011, p. 1; see also Dolan 1998, p. 66; Hill 2007, pp. 100–01). The Irish servant was, therefore, just as in Britain, a gothic monster, an archetype of primitive savagery from across the Atlantic, invading that most sacred sanctuary of safety and security—the middle-class American home.

The stereotype of "Bridget," the inept, coarse-featured, insubordinate Irish servant, was, moreover, a "grotesque inversion" of "nineteenth-century feminine and domestic norms" (Flynn 2011, pp. 1–2). Nevertheless, this opposition became more difficult to maintain as the century wore on; domestic service was in fact seen as a direct route to middle-class status for white women, facilitating Irish modernization, assimilation, and "socialization" (Katzman 1981, pp. 156, 171; Sutherland 1981, p. 27; Dudden 1983, pp. 24, 167; O'Neill 2009, p. 115; Flynn 2011, p. 19). By the turn of the century, the Irish servant girl was transformed into the epitome of the American spirit of independence, adaptability, hard work, and self-sacrifice (Groneman 1978, p. 258; Murphy 1998, pp. 133–35; Dezell 2001, pp. 91–93; Howes 2009, p. 99). From being "the inassimilable alien," the Irish servant became "virtually indistinguishable from other female members of the household," as she learned how to look, dress, behave, talk, and clean house in the "American" way (Murphy 2000, p. 172). This in itself helped sharpen American Protestant discomfort, however, and Irish women's integration into American society was initially met with derision, as satirical cartoons portrayed Irish "Bridgets" as "simian-faced ingrates in fancy dresses who have trespassed well above their station" (O'Neill 2009, p. 126; see also Murphy 2000, p. 153; Dezell 2001, p. 19). Like many young female domestic workers enjoying the power of disposable income, Irish servant girls liked fine clothes (Katzman 1981, p. 237; Sutherland 1981, pp. 30, 127; Dudden 1983, p. 120; Lynch-Brennan 2009, p. 76; Turner 2012, p. 200). Although they were frequently lampooned for poor taste, this propensity caused potential disruption to the visual register that had hitherto signaled distinctions between classes. In the process, the Irish servant girl began uncomfortably to resemble the "mistress" of the house, while remaining haunted by the specter of her supposedly barbarous heritage. Figuratively, the Irish servant girl was therefore pushed in two directions at once—towards the future and the past, middle-class respectability and peasant superstition—and derided for both.

## 4. Martin and Bergstrom's Irish Servants

This figurative tension is dramatized in *Mary Reilly* and *Blood to Blood* through doubling, as Mary and Colleen gradually become extensions or facsimiles of their employers, while remaining shadowed by implicitly Irish forms of monstrosity. What this means, I argue, is that, even as they inhabit the role of gothic heroine, bringing light to dark places and undermining patriarchal tyranny, they also become associated with the irrational forces of gothic monstrosity (specifically vampirism) that heroines are tasked with exorcizing. Indeed, their role as servants is equally responsible for their

impulse to clean and illuminate, and for their positioning as doubles of their employers. Katzman (1981) argues that, because they spent even more time in the house than their employers, housemaids were closely attuned to domestic atmospheres, and at the mercy of the moods and troubles that employers brought home with them, making them good barometers of the moral and emotional health of a household (pp. 270–71). According to Flynn (2011), this sensitivity found expression in later, still-stereotyped but far more flattering depictions of the Irish servant girl in America, which depicted her as "imposing herself upon her employers not as an unruly servant, but as an equal" (p. 14) as she progressively attains American manners and sensibilities. This is especially notable in the 1917 film *Pots-and-Pans Peggy*, in which an Irish servant girl "cleans up" the laziness and poor romantic choices of her employer-family, showing herself to be their moral and even intellectual superior. Significantly, however, the film implies that this superiority is best employed not in improving her own circumstances, but in the service of "fixing" Peggy's Protestant, middle-class employers—that is, in the performance of precisely the labor undertaken by the gothic heroine, uncovering abuses within the home so that they may be rejected and surmounted (p. 17). However, like Mary Reilly, and unlike conventional gothic heroines, she does this not for her own sake, but for theirs.

The novels discussed in this article retroactively engage with the contradictions inherent in this position for Irish servants, literally and figuratively cleaning house for a social class that they can never fully join. Martin's Mary Reilly seeks to encourage Dr Jekyll back into the domestic realm, away from London's backstreets and Hyde's pernicious influence, while Colleen, in *Blood to Blood*, works hard to assimilate Joanna Tepes, Vlad's Turkish-born adopted sister, into Western society, teaching her how to avoid social and legal censure in an urban world of surveillance and "proper" feminine behavior. However, it is Colleen's insubordination that allows her to approximate the role of gothic heroine, revealing male violence that employs social privilege as a smokescreen. Colleen is therefore central to the successful re-appropriation of the role, usurped by Stoker's male characters. Refusing to sit quietly at home while Joanna goes out hunting at night is what allows Colleen to learn that Dr Rhys (a charming physician who develops an interest in Mina's charitable efforts) is an analogue of Jack the Ripper. Following Dr Rhys through the streets of London provides her with proof of his extracurricular activities, but also, in true gothic-heroine fashion, places her in danger; he notices her and attacks. However, this hastens her complete transformation into a vampire, ensuring that later, when Dr Rhys attempts to kill Mina and her servant Essie, Colleen is strong enough to fight him, giving Mina an opportunity to shoot him dead.

On a number of levels, this depiction of Colleen adheres closely to nineteenth-century American stereotypes of Irish servant girls, who were often "characterized [ . . . ] as impertinent and assertive," accustomed as they were to "a certain broader latitude, in terms of acceptable female behavior [ . . . ] than was allowed middle-class women in urban America" (Lynch-Brennan 2009, p. 35). Irish servant girls were far more self-confident than their employers expected girls of their age and station to be, and did not simply take orders meekly and unquestioningly. These earlier associations are easily identifiable as persisting in Bergstrom's novel; indeed, it is notable that Essie, Mina's servant, who appears to be English-born, rarely questions or resists her mistress's commands. Significantly, however, this contrast does not hold true in relation to superstition and knowledge of the supernatural. On the ship to England from Varna, soon after they first meet, Colleen assures her new mistress that she recognizes her for what she is, and that she can attend without fear to the vampire's "'unique needs,'" referencing her awareness of the existence of "'[b]anshees. Dearg-dul. Rakashas'" (Bergstrom 2000, p. 51), a list that combines Irish and Hindu supernatural entities (and warrants additional exploration). Essie's cultural references are less cosmopolitan, but she is equally calm when Mina tells her that vampires are real, reassuring her employer that she has read *Varney the Vampire* (Rymer and Prest [1847] 1972) and other "'old books'" (Bergstrom 2000, p. 85). Indeed, despite her implicit Protestantism, and unlike the men in Stoker's novel, Essie is perfectly happy to use holy water, a gold cross, and a bell (suggesting the "bell, book, and candle" of exorcism) as safeguards against undead visitors. Working-class women, *Blood to Blood* implies, experience far less cognitive dissonance than their social

"superiors" when confronted with the otherworldly, and are therefore of great use to both skeptical middle-class employers and monstrous aristocrats.

At the same time, vampirism facilitates the erosion of social and psychic differences between Colleen and her mistress. When Joanna disappears on an extended hunt, leaving an elaborate green dress for Colleen, we are told that "[i]t never occurred to her where an Irish working girl already living well beyond her class would be able to wear it" (Bergstrom 2000, p. 231). Devastated at being abandoned for so long, Colleen wonders if "her ordeal was even some sort of test to determine if she were worthy to become one of those nocturnal creatures" (p. 233), after which she decides to wear the dress, which allows her to pass as a woman of a higher class, suggesting that she is in fact "worthy" to ascend to the status of aristocratic vampire. Her gradual supernatural transformation therefore parallels her fluctuating class position; indeed, she describes herself as "*between two worlds*" (p. 235, italics in original)—both socially and ontologically. Nevertheless, even when this liminal position allows Colleen to save the day, Joanna is still referred to as her "mistress" (p. 304). While she may be permitted some narrative space in which to be a sleuthing heroine, then, and is temporarily permitted to "rise" socially, the Irish servant is relegated to the position of a mere enabler of her aristocratic employer; Victorian class divisions may be briefly unsettled, but remain largely intact. At the same time, for Mina (whose own middle-class respectability has been sufficiently compromised by her encounter with Dracula that, in *Mina*, she has an affair with the sinister Lord Gance, and then leaves Jonathan, moving into a house that her wealthy lover bequeaths her), the vampires are no longer monstrous. Rejecting Van Helsing's dogma, which insists that vampires cannot control their appetites and retain no human qualities, by the end of *Blood to Blood*, she and Essie (and even Jonathan, with whom she is ultimately reconciled) are happy to have Joanna and Colleen haunting their garden, an arrangement that presumably also suits Arthur, whose sexual proclivities now strongly favor the undead. The book signals its progressive tendencies through the cheerful transgression of previously strict sexual and ontological boundaries, but class boundaries remain more intransigent, it seems.

It is therefore notable that Martin's book, which is more constrained by the plot of Stevenson's novel than Bergstrom's is by Stoker's, presents a far more rigid fictional universe, in which class and species boundaries are strictly maintained in the face of the threat represented by Hyde, who is here also associated with vampiric blood sucking and infection. Bryk (2004) asserts that, "rather than mounting a challenge to the precursor text, [Martin] reproduces [*Jekyll and Hyde*] with an almost reverential faithfulness" (p. 206) that is matched by Mary's reverential faithfulness towards Jekyll—a faith so trusting that she rarely questions his motives enough to understand what is happening to him, despite her concerns. She repeatedly tells herself to remember her "place" in society and in the Jekyll household (particularly when she finds herself thinking of him as anything other than an exemplar of moral rectitude), and even her distant "Master" notes that she has "'a fairly profound view of social order and propriety'" (Martin 2003, p. 16). It is therefore unsurprising that Mary conforms less closely than Colleen to the stereotype of the rebellious Irish maidservant, and indeed, whether she is Irish at all is very much up for debate. McFadden (2011) asserts that "[w]hile Irishness is suggested in the surname Reilly, there is no mention of Mary's Irish identity or ancestry in the three diary volumes that comprise Martin's novel. The 'voice' Martin grants Mary is a soundless voice and Mary's accent is undetectable in the written word" (pp. 65–66). This is perhaps stating things too strongly; elements of direct translation from Irish into English appear in the grammar of her first-person narrative. Early on in *Mary Reilly*, Jekyll asks to see the scars on Mary's hands (inflicted by her violent father) while she is cleaning the grate, and she tells him that she is worried about soiling his clothes with soot, insisting that "'*it do travel* no matter how I might try'" (Martin 2003, p. 6, italics added). This bears some resemblance to the Hiberno-English construction "it do be," from phrases such as "*bíonn sé ag taisteal*" ("it do be travelling/ it tends to travel"). Mary also occasionally employs a gerund where one might expect the present tense, reflecting a pattern of usage in Irish. For example, she remarks, "I thought it odd that Master *would be running* a school he never saw" (p. 29, italics added). She also, however, repeatedly uses the Northern-English word "mum" for "must" (see p. 243). Her Irishness

is simultaneously gestured toward and undermined by her vernacular. Mary is perhaps therefore legible as second- or third-generation London Irish, rendering her a less threatening figure than an unambiguously Irish character might have been in the world of *fin-de-siècle* (Scottish-)Protestant gothic. Moreover, her narration contains fewer of these vague dialect markers in the second half of the novel, implying a growing identification with the class she serves.

This integration into middle-class London society, to the extent that her Irish heritage remains only as a barely identifiable linguistic and allegorical haunting, prevents her from doing her job—from "cleaning" her Master's house figuratively as well as morally—because she is also tasked with keeping his secrets, and therefore fails or refuses to understand the relationship between Jekyll and Hyde until it is too late. Indeed, for most of the novel, she almost literally cannot or will not see at all, and, unlike Colleen, she therefore never fully reclaims the gothic-heroine role from Utterson and Lanyon (see Bryk 2004, p. 207 for a contrary reading). The morning after she surprises Hyde writing in Jekyll's check book, her Master asks Mary to bring it to him, saying (to cover for Hyde) that he had failed to complete an entry in it. Mary begins to question rather than simply obey, but doing so "made my hands tremble; I longed to open the book and see for myself what was there, but I hadn't the nerve" (Martin 2003, p. 99). She is too afraid of what might be revealed about Jekyll to act the part of gothic heroine, and too unwilling to stray beyond the exact limits of her duties to save her Master from himself.

This is acknowledged near the end of the book, when Mary finds Hyde's dead body in Jekyll's private cabinet and finally understands the connection between them; she "set the lamp upon the table where it made a great clatter of light among the bottles [ ... ]. All the time the truth was right before my eyes [ ... ] but I would not understand, as if I was too stubborn to know it" (p. 236)—and, more importantly, too late to do anything about it. This great "clatter of light" (another vaguely Hiberno-English expression) may now illuminate the situation, but Mary has not succeeded, as Kiraly/Bergstrom's Mina does, in extricating herself from the predetermined textual pattern against which she struggles, nor does she bring male violence and power into the light of day, as Colleen does. However, like Colleen, she remains a secondary character, despite her usurpation of the narrative voice from Jekyll and his male friends. The drama of the narrative remains Jekyll's; her role is merely interpretive—and she fails even at that. Mary is therefore too well assimilated to perform the full duties of the gothic heroine, just as the book itself leaves the plot and the ideological structures of Stevenson's original largely intact. If anything, she is complicit in what happens, as her Master entrusts her with private errands to the extent that she becomes an extension of his will. The conditions of service in nineteenth-century Britain and America tended to transform servants into virtual extensions of their employer (Blackford 2005, pp. 235–36; see also Katzman 1981, pp. 159, 236, 269; Dudden 1983, p. 115), with little agency or individual identity of their own (Thompson 2015, pp. 93–94; Pereen 2014, p. 78). As Hyde makes life ever more difficult for Jekyll, Mary is transformed (albeit reluctantly) into an agent of Jekyll's efforts to cover up his double's terrible deeds. When Jekyll asks her to deliver a letter for him to a Mrs Farraday (who turns out to own a brothel), she is "left sleepless, feeling not trusted and valued as I should, but anxious and afraid" (Martin 2003, p. 56), realizing that his request positions her as an accomplice to Hyde's crimes, rather than as a gothic heroine investigating them. Later, when Jekyll sends her back with a check (to help conceal the fact that one of the girls seems to have been murdered by Hyde), she comments, "I knew at once why he had sent for me. It was to carry his anger out of the house," recognizing that she acts as his "hands" (p. 110). Her job, therefore, is housecleaning, but for the purposes of evacuation and concealment rather than revelation. She, like Hyde, "carries his anger out of the house" so that it can remain outwardly pure—an agent of the patriarchal power and sexual hypocrisy that the book associates with Jekyll and his friends, apparent pillars of the community who frequent burlesque shows and are acquainted with brothel owners in Soho.

Mary's alignment with Hyde is rendered more explicit in the film adaptation, in which Mary, played by Julia Roberts, attempts a rough approximation of an Irish accent, rendering audible the novel's textual ambiguity (McFadden 2011, pp. 65–66). Mary's alcoholic father is depicted as Irish by Michael Gambon (who was born in Ireland, but moved to London at the age of five); and Hyde (played by

John Malkovich, who also plays Jekyll) occasionally speaks with a "distracting" Irish accent (p. 70). What this confluence of implied ethnicity helps highlight retrospectively in Martin's novel is the extent to which Hyde and Mary's father (who haunts her memory as Hyde haunts Jekyll) are figuratively connected. When Mary first hears Hyde coming up the back stairs to Jekyll's bedroom late at night, his tread sounds exactly like the "halting," "dragging" step of her father coming to abuse her (Martin 2003, pp. 33–35, 89), an association made numerous times throughout the novel. A psychological reading might see this as implying that Hyde is a mere projection of her childhood trauma at her father's hands. In light of what I have been saying about *Dracula* and *Jekyll and Hyde*'s connections with Irish stereotypes, however, it can be seen as directly linking Mary's heritage to Hyde's supposedly degenerate "deformity" (Stevenson [1886] 2008, p. 9).

Indeed, Hyde serves to connect Mary to nineteenth-century discourses regarding Irish assimilation in both Britain and the United States, and specifically to fears regarding social mobility. Hyde, who is clearly not a "gentleman" (Martin 2003, p. 106), has been elevated above her socially and acts as if she must serve him, coming in the front door (p. 104) like a guest or a full member of the household. If we accept the reading that Hyde metonymizes the dangerous-yet-necessary Irish immigrant in Victorian London, and the interpretation of Mary as a "naturalized" child of Irish parents or grandparents, then Hyde acts as her shadow, a reminder that her heritage carries distinctly gothic connotations, both in Victorian Britain and in American cultural memory. Indeed, in the text itself, he singles her out from the rest of the household servants, seeming to harbor a particular resentment towards her, and then effectively attempts to contaminate her with his monstrous Irish mobility, just as Dracula contaminates Lucy and Mina, and as she fears to contaminate Jekyll with soot, a signifier of her lowly occupation. In the first of two sanguinary encounters between Hyde and Mary, he cuts his hand on a broken cup after insisting that she serve him tea, and tries to force her to drink his blood (p. 148). In the second, she realizes that he sees her as "prey" (p. 217), and he bites her neck deeply, but, importantly, without managing to draw blood (p. 220). As in *Blood to Blood*, then, vampirism and social mobility are connected here, but in this case, Mary resists the temptation where Colleen succumbs—or at least, Hyde fails to penetrate her skin and identity. Concomitantly, where Colleen's (implicitly Catholic) knowledge of the supernatural helps save the lives of Joanna, Mina, and Essie, Mary's refusal of that knowledge means that she can never fully assume the gothic heroine's light-bringing, villain-thwarting duties.

## 5. Conclusions

Martin and Bergstrom's novels therefore neatly combine two of the key figurative uses of the vampire, laminating onto one another fears surrounding class difference and infection (Wasson and Alder 2011, p. 5). As Stewart (1999) notes, Stoker's "Dracula represents *both* the landlord *and* the forces" of what was perceived to be "atavistic violence"—specifically, late-nineteenth-century Fenian activism (pp. 240–43, italics in original). In other words, the vampire simultaneously connotes the blood-sucking avarice of the rich (Deane 1997, p. 90; Eagleton 1995, p. 215) and the dangerous mob mentality of the oppressed. Understood within the context of Irish mobility in the nineteenth century, Martin's Hyde represents both the working-class Irish identity from which Mary seeks to extricate herself, and the violent privilege represented by Jekyll and his boorish, snobbish, misogynistic friends. In addition, as Sage (1988) points out, the Catholic practice of taking the Eucharist at mass, eating and drinking what is seen as the actual body and blood of Christ, was often figured as itself an act of vampirism by Protestant commentators (pp. 50–56). In other words, the vampiric threat embodied by Hyde is a threat to Mary's precarious respectability, a respectability that demands a rejection of her implicitly Irish-Catholic heritage.

This threat is dramatized as liberating in Bergstrom's novel, opening up new social and sexual possibilities for Mina and those around her. Nevertheless, Colleen, while a part of Mina's newly widened and tolerant social circle, remains socially and narratively subordinate to both her mistress and the middle-class characters. She, like Mary, ultimately knows her place, even if she saves the day by briefly stepping outside of it and by embracing the monstrosity (also figuratively associated with

Irishness and Catholicism) that Mary rejects and that never directly threatens the English-born Essie. Both characters are therefore caught between class identities, but it is Mary, the more conservative character in the more conservative text, who finds herself helplessly doubled, a prosthetic extension of her Master's will, and in uncomfortable proximity to Hyde's ethnic and class affiliations, with which he threatens to infect her through blood. This infection has positive outcomes for Colleen, preserving her life and allowing her to rescue her social "superiors" from a very human serial killer; but while Mary seems to survive the events of the novel, Jekyll does not. Her failure or refusal to save him by grasping the truth about his relationship to Hyde can perhaps be attributed to her insistence on distancing herself from the supernatural aspects of Irish-Catholic identity, especially in light of the successful integration of the Irish into twentieth-century American society (Katzman 1981, pp. 121–122, 160; Dolan 2008, p. 305). Indeed, in the latter half of the twentieth century, the ideal Irish traits came to be seen as the ideal American traits (Dezell 2001, pp. 24–25); specifically, Irish immigrants' "gregariousness, wit, charm" (Dolan 2008, p. 307) helped transform them into "America's favorite ethnics" (McCaffrey 2000, p. 18). Integrated the Irish might be, but they remain marked by a particular set of ethnic associations, even if those associations have gone from menacing American middle-class identity to bolstering it.

A contemporaneous text, the television series *Angel* (1999–2004), neatly exemplifies the pattern that I have outlined here. The hero of the show, Angel (originally Liam), hails from a wealthy merchant family in eighteenth-century Galway (Meaney 2006, p. 263), the religious affiliations of which are unclear. Angel is now a (reformed) vampire who, in line with the vampire-landlord trope, consistently inhabits large, imposing buildings; even in America, he has never been poor, and so is only tangentially associated with nineteenth-century stereotypes. In Season 1 (1999–2000), however, his assistant in his efforts to fight crime and otherworldly evil in modern-day Los Angeles is Francis Doyle, a charming young man from Dublin with a checkered past, a love of drink, and a mountain of debt. Like Colleen, Doyle comes equipped with valuable knowledge of the supernatural underground, and, half-demon, has some useful powers of his own. Like Mary and Colleen, Doyle remains a secondary character, and uses his visions of impending danger to help his employer, who is repeatedly figured as better looking and financially better off than his more stereotypical sidekick. The class differences between the two men cannot, it seems, ever be overcome. What this suggests, I would argue, is that, as Martin and Bergstrom's novels make clear, the working-class Irish in the American imagination may have been transformed from clumsy savages to loveable rogues, but their longstanding associations with antiquity, monstrosity, superstition, and servility continue to serve a very distinct imaginative purpose. Shambling and strutting like Martin's Hyde, the working Irish embody a gothic past that resists exorcism.

**Funding:** This research received no external funding.

**Acknowledgments:** This paper was partially researched during a one-week Moore Institute Fellowship in the National University of Ireland, Galway, in April 2018.

**Conflicts of Interest:** The authors declare no conflict of interest.

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
