# Peer review of "The “Irish” Female Servant in Valerie Martin’s Mary Reilly and Elaine Bergstrom’s Blood to Blood"

_humanities, doi:10.3390/h9040128_

Round 1

Reviewer 1 Report

This is a strong, well-researched essay which merits publication. The discussion of the contemporary texts is largely convincing - while the suggestion of Atwood and Sheridan at the text's conclusion is tantalising, the argument is well-constructed with its current focus. My suggestions, which should certainly only be seen as minor corrections given the limitations of space, are more focused on some of the contextual material. More consideration could be given to the lineage of, and difference between, Irish and Scottish Gothic. While the author supports the reading of Hyde as an Irish figure well, there's an opposed reading of the novel that focuses more on Jekyll's relationship with his father and the more complex view of Scottish Protestant culture that reveals. This is a relatively small point, but simply to suggest that British Protestantism is not particularly unified, and that many of RLS's themes relate very specifically to a Scottish context. More importantly, the readings of Stoker's work in an Irish context are a bit light - there may be a reason for not citing Terry Eagleton's work here, but I'm surprised not to see it. Likewise, there are a few key touchstones for work on Irish/Scottish Gothic and religion that I might expect to see footnoted - Victor Sage, Crawford Gribben or, more recently, Neil Syme's chapter in Eleanor Beal and Jonathan Greenaway's volume on Horror and Religion. Certainly not all of this can be included, but I think the article might do more with the fact that both of the source texts are set (mostly) in England, are not by English authors, and are often read as reflecting that difference - which is all a protracted way of saying that the jumps on page 5 might use a little unpacking. I think, too, just a line or two discussing the association in the late nineteenth century between Irish emigration and Frankenstein would help solidify the argument. Once the article is into the main body I think it works very well. The one area that needs a little more development, perhaps, is how the stereotypes that are well-described in the article might resonate with contemporary readers (and if the contemporary texts themselves have different readerships with different expectations). The conclusion is a bit rushed, and while it does point promisingly to further avenues of enquiry, I'd like to see them developed a bit more here.

Again, I accept that not all of these points can be considered, and this is a rich article already. The essay is well-written and needs only a small amount of editing (the bibliography does have some ordering problems). This is a well-researched and insightful piece - my strongest recommendation would be for the author to consider a little bit more how the tropes they examine extend both backwards and forwards from the texts discussed.

Reviewer 2 Report

This article offers a reading of two neo-Victorian gothic novels as a lens through which to view fictional representations of Irishness. The central premise is that while marginalised figures are made more prominent in the two novels, their class / immigrant status is insufficiently problematised. The article is predicated on the gender of the respective protagonists, but highlights the ways in which revisionary texts can themselves fail in their own (implicitly feminist) agenda, as they adhere to unexamined assumptions based on the social position of Irish domestic workers.

This is an intriguing argument but could be further nuanced in the introduction. The point that ‘Reading Mary Reilly in tandem with Blood to Blood helps highlight the ways in which Irish servants on both sides of the Atlantic have long been associated with a gothicized European past’ (56-8) could be made more explicitly in the opening paragraph. It felt rather as if this point was being first taken as read and subsequently investigated, although in one sense this is a testament to the author's own knowledge of the source material! 

The article offers a comprehensive review of existing criticism, but in the first half the author's own voice is in danger of being submerged by it. A brief synopsis of the two novels would be very welcome early on.

The article would also benefit from a less divided approach - the context and historic background takes up around 8 pages, leaving little room for sustained analysis of the novels. However the analysis of Mary Reilly's Irish markers is notably insightful, as is the discussion of Irish workers being variously assimilated into the American class system and acting as markers of degeneration.

This article shows a strong command of the material and a closer alignment of context with a more detailed literary analysis would make for a really compelling argument.

Round 2

Reviewer 2 Report

I agree with reviewer 1 that this article has a strong case to make, but I think a more sustained analysis of the fiction would have lifted it into another league. The argument is slightly hard to follow in places because the novels themselves seem secondary to the wider historical agenda, hence my suggestion that a synopsis of the texts should appear early on. I have revisited the synopsis of Blood to Blood and it did seem clearer on a second reading. But I still struggled to work out the basic plot of Mary Reilly, beyond its focus on the perspective of a marginal figure from the original source text. The key elements are covered in the course of the essay, but the reader is being asked to do additional work here. A comparatively small amount of reorganisation and additional explanation could have made a huge difference. 

I would still recommend publication based on the revisions, although I got the impression that the author was slightly resistant to rebalancing the context / literary analysis. Another round of revisions would have been well worth the effort. nb track changes can always be taken out for easier reading, but in any case this would not be a valid reason for not making revisions!